# The Pathological Role of miRNAs in Endometriosis

**DOI:** 10.3390/biomedicines11113087

**Published:** 2023-11-17

**Authors:** Mst Ismat Ara Begum, Lin Chuan, Seong-Tshool Hong, Hee-Suk Chae

**Affiliations:** 1Department of Biomedical Sciences, Institute for Medical Science, Jeonbuk National University Medical School, Jeonju 54907, Republic of Korea; ismatara1986@gmail.com (M.I.A.B.); mumuli415@163.com (L.C.); 2Department of Obstetrics and Gynecology, Research Institute of Clinical Medicine, Jeonbuk National University, Jeonju 54907, Republic of Korea; 3Biomedical Research Institute, Jeonbuk National University Hospital, Jeonbuk National University Medical School, Jeonju 54907, Republic of Korea

**Keywords:** miRNA, endometriosis, endometriotic lesions, infertility, eutopic endometrium, diagnostic markers, therapeutic perspective

## Abstract

Association studies investigating miRNA in relation to diseases have consistently shown significant alterations in miRNA expression, particularly within inflammatory pathways, where they regulate inflammatory cytokines, transcription factors (such as NF-κB, STAT3, HIF1α), and inflammatory proteins (including COX-2 and iNOS). Given that endometriosis (EMS) is characterized as an inflammatory disease, albeit one influenced by estrogen levels, it is natural to speculate about the connection between EMS and miRNA. Recent research has indeed confirmed alterations in the expression levels of numerous microRNAs (miRNAs) in both endometriotic lesions and the eutopic endometrium of women with EMS, when compared to healthy controls. The undeniable association of miRNAs with EMS hints at the emergence of a new era in the study of miRNA in the context of EMS. This article reviews the advancements made in understanding the pathological role of miRNA in EMS and its association with EMS-associated infertility. These findings contribute to the ongoing pursuit of developing miRNA-based therapeutics and diagnostic markers for EMS.

## 1. Introduction

Endometriosis (EMS), characterized by the presence of endometrial stromal and glandular tissue outside the uterine cavity, is a relatively common gynecologic disease affecting about 10% of women of reproductive age [1]. While the symptoms of EMS can vary, clinical manifestations such as pelvic pain, heavy or irregular menstrual bleeding, and gastrointestinal symptoms are frequently observed. Apart from the physical discomfort, EMS has been well noticed in infertility. It is noteworthy that approximately 10% of women of reproductive age are affected by EMS, and a striking 30–50% of women with EMS experience infertility, highlighting the significant role this condition plays in infertility [2]. Since severe adhesions can distort the pelvic anatomical structure in cases of severe EMS, the role of EMS in causing infertility can be easily explained. These anatomical abnormalities are not typically observed in cases of minimal and mild EMS, but a high frequency of infertility is still noted [3]. Surgical treatment with a laparoscopy in women with minimal or mild EMS increases their pregnancy rate to 30.7% from 17.7% in the non-treated group over a 20-week observational period [3,4]. Despite this increase, the pregnancy rates after surgical treatment in women with minimal or mild EMS are lower than those in normal fertile women, suggesting an association between EMS and infertility beyond anatomical abnormalities [5]. Mechanisms such as abnormal folliculogenesis [6], elevated oxidative stress [7], altered immune functions [8], and hormonal milieu in the follicular and peritoneal environment [9], and reduced endometrial receptivity [10] have been proposed to account for fertility impairment in women with EMS, especially mild EMS without anatomical abnormalities due to adhesion, but there is still no satisfactory explanation [11,12].

Considering the severity and prevalence, as well as the lack of solutions for EMS, the elucidation of the etiopathogenic mechanism of EMS becomes one of the critical issues in modern medicine. Recent studies have shown that microRNAs (miRNAs) play significant roles in the etiopathogenesis of diseases along with proteins. microRNfAs are small non-coding RNAs, approximately 21–22 nucleotides in length, known to serve an important function as modulators of gene expression by targeting mRNA for degradation or translation repression [13]. This post-transcriptional regulation of gene expression occurs both in physiological conditions, including cell growth, development, differentiation, proliferation, and cell death, and in pathological conditions, such as cancer and inflammatory diseases [14,15].

Despite the significant role of miRNAs in the pathogenesis of diseases, their importance has not received enough attention in medical research. This review aims to update the current knowledge on the pathogenic role of miRNA in EMS and EMS-associated infertility, as well as the possibility of using miRNA as miRNA-based therapeutics and diagnostic markers for EMS.

## 2. The Biology of miRNA

miRNAs are small, single-stranded, non-coding RNAs, averaging 22 nucleotides in length. These RNA molecules are highly conserved across eukaryotic species and do not encode proteins. Instead, they modulate post-transcriptional gene expression by binding to the untranslated regions (UTRs) of mRNA at either the 3′ or 5′ terminal ends, preventing protein production. Their critical role in gene regulation is underscored by the fact that approximately one-third of human gene expression is under the control of miRNA [16,17]. Despite their inability to produce proteins, miRNAs exert significant influence over a wide range of biological processes by participating in crucial pathways and contributing to the regulation of normal animal development [18]. The miRNA database (miRBase) has expanded its catalog, listing 434 miRNAs in *Caenorhabditis elegans*, 466 in *Drosophila melanogaster*, and a substantial 2588 in humans [19]. miRNA genes are located throughout the genome, spanning functional gene coding and noncoding regions. These genes can be expressed either in clusters or independently.

miRNA is synthesized by either RNA polymerase II or III, followed by the processing of RNA transcripts either post- or co-transcriptionally [20]. This process of miRNA biogenesis is categorized into canonical and noncanonical pathways. The canonical biogenesis pathway (Figure 1) is the dominant route through which miRNAs undergo processing. The biogenesis of miRNAs via the canonical pathway can be summarized in five key stages [21,22,23,24,25], described subsequently:

① Transcription: Canonical primary miRNAs are transcribed by polymerase II (although polymerase III-transcribed miRNA precursors are uncommon) to yield primary miRNA (pri-miRNA) transcripts. The pri-miRNA transcripts can arise within protein-coding genes or intergenic regions.

② DROSHA-Mediated Cleavage: The Drosha nuclease, in conjunction with the RNA-binding protein DGCR8 (forming the Microprocessor complex), initiates endoribonucleolytic cleavage, excising the 5′ and 3′ ends of the pri-miRNA. The outcome is a short hairpin structure known as pre-microRNA (pre-miRNA), typically comprising 60 to 70 nucleotides.

③ Nuclear Export via Exportin-5: Exportin-5, in association with Ran and GTP, binds to the pre-miRNA. This complex facilitates the movement of pre-miRNA through the nuclear pore, translocating it to the cytoplasm.

④ Dicer-Driven Cleavage: within the cytoplasm, Dicer, another enzyme, cleaves the pre-miRNA, generating a double-stranded miRNA duplex of approximately 22 bp in length.

⑤ Incorporation into the RNA-Induced Silencing Complex (RISC) and Strand Selection: one strand of the miRNA duplex, designated as the guide strand, is preferentially chosen to serve as the mature functional miRNA, while the passenger strand is usually degraded.

In addition, beyond the canonical route, several non-canonical miRNA biogenesis pathways have been unveiled. While these pathways exhibit structural and functional similarities to canonical miRNAs, they diverge in their maturation processes, often circumventing specific steps of the conventional biogenesis pathway [26]. These non-canonical pathways employ varying combinations of key proteins found in the canonical pathway, predominantly including Drosha, Dicer, exportin 5, and AGO2 [27]. Broadly, non-canonical miRNA biogenesis can be categorized into two groups: those that are independent of Drosha/DGCR8 and those that are independent of Dicer. The Drosha/DGCR8-independent pathway generates pre-miRNAs resembling Dicer substrates. Notably, “mirtrons” exemplify this type of pre-miRNA, originating from mRNA introns during splicing [28,29].

## 3. The Pathological Role of miRNA in Endometriosis

EMS is a common gynecological disorder characterized by pelvic pain, heavy menstruation, and infertility. Roughly half of women affected by EMS encounter difficulties with infertility [30,31]. Multiple studies indicate that women in the early stages of EMS face reduced prospects of natural conception, and there has been a growing prevalence of EMS among women experiencing infertility [5,32]. Among patients with this milder form of the disease, potential mechanisms leading to infertility include persistent inflammation, disrupted ovulation, impaired embryo implantation, and suboptimal receptivity of the endometrial lining [33]. Nonetheless, the precise underlying mechanism of infertility linked to EMS remains elusive.

Although the etiopathogenic mechanisms of EMS have not yet been fully elucidated, the current research indicates that miRNAs are involved in the disease. The differential expression of miRNAs has been repeatedly observed in endometrial tissues and in the extracellular body fluids of women with EMS in contrast to those without the condition [12,34]. This variance in miRNA expression signifies disrupted gene regulation. Profiling miRNAs in individuals with EMS holds the potential to provide valuable insights into understanding the mechanisms driving the development of the disease.

### 3.1. The Pathogenic Mechanisms of Endometriosis

While EMS is often linked to infertility [35], the precise mechanisms driving this association remain incompletely understood. Numerous investigations have been undertaken to shed light on this issue, with various authors proposing diverse mechanisms that could potentially contribute to infertility [36,37,38]. These mechanisms encompass anatomical and microenvironmental factors that may have detrimental effects on oocyte maturation, fertilization, zygote transportation within the fallopian tube, embryo implantation, and even sperms function.

#### 3.1.1. Ovary Dysfunction

Patients with EMS exhibited the abnormal secretion of Luteinizing Hormone (LH), potentially leading to ovarian dysfunction [39]. Alterations in folliculogenesis among individuals with EMS may contribute to issues such as ovulatory dysfunction, diminished oocyte quality, reduced fertilization rates, the production of lower-grade embryos, and decreased implantation success [6,40]. In an observational in vitro fertilization (IVF) study involving natural cycles, it was observed that patients with minimal-to-mild EMS had significantly longer follicular phases and lower fertilization rates when compared to women with tubal factor infertility and unexplained infertility [39]. Women with EMS were also found to exhibit a slower rate of follicular growth [41] and smaller dominant follicles compared to women with unexplained infertility [42]. Additionally, EMS patients often experience abnormal follicle development, ovulation, and luteal function [43].

#### 3.1.2. Dysregulated Immune Function, Hormonal Imbalance, and Oxygen Species in Follicular Fluid

EMS is linked to inflammatory changes in the follicular fluid (FF) environment. A case-controlled study comparing patients with EMS to those with other causes of infertility has noted an increased percentage of B lymphocytes, Natural Killer (NK) cells, and monocyte–macrophages in the FF. This suggests potential alterations in the immunological function within the FF of individuals with EMS [44]. Elevated concentrations of interleukins IL-6, IL-1b, IL-10, and Tumor Necrosis Factor-alpha (TNF-alpha), along with decreased Vascular Endothelial Growth Factor (VEGF), have been documented in the FF of EMS patients [45,46,47]. Immunological changes in the follicular fluid (FF) and serum of women with EMS may contribute, at least in part, to the pathological changes associated with infertility in these patients [8]. For instance, VEGF, known for enhancing follicular health and vascularization, is found in reduced levels in women with EMS [46], potentially associated with diminished embryo quality and implantation rates [6].

Notably, significantly elevated concentrations of TNF-alpha in granulosa cell cultures from women with EMS have been reported [48], which may also be linked to infertility [49]. A study demonstrated that the use of etanercept, an immunoglobulin fusion protein that hinders TNF-alpha activity, reduced the severity of EMS and the size of endometriotic foci in an animal study [50]. Unlike the effect on EMS, the inhibition of TNF-α did not appear to affect infertility [51]. However, the use of etanercept before in vitro fertilization (IVF) enhanced the success rate in cases of advanced EMS [51]. Cytokine changes may influence alterations in the granulosa cell cycle, as previously observed in EMS patients [7]. For example, elevated levels of IL-10 were shown to prevent the natural downregulation of p27, leading to a G0 phase arrest [52]. Several other cytokines, elevated in the FF of EMS patients, such as IL-6, IL-1b, IL-8, or IL-1a, likely contribute to various cell cycle abnormalities, further contributing to subfertility in these patients [7].

Additionally, increased levels of IL-6 in the preovulatory follicles of EMS patients have been found to result in decreased aromatase activity via the MAPK signal pathway. This reduced aromatase activity leads to the decreased intrafollicular conversion of androstenedione to estrone and the subsequently diminished conversion of androstenedione to testosterone, which is aromatized to estradiol (E2) [53,54]. This decrease in follicular E2 levels may result in fertility issues, including reduced fertilizing capacity [53]. Changes in progesterone levels have also been observed in the FF of EMS patients, suggesting that altered steroidogenesis likely plays a significant role in EMS-associated infertility. However, a direct relationship between infertility and modified progesterone levels has yet to be firmly established [6]. Some researchers have proposed impaired LH production as the primary pathophysiology underlying impaired ovulation [39]. It has been suggested that Gonadotropin-Surge Attenuating Factor (GnSAF), a small polypeptide primarily produced by small follicles in the FF, contributes to decreased LH levels in EMS patients. GnSAF decreases the ability of E2 to sensitize the pituitary to the gonadotropin-releasing hormone, thereby reducing the pituitary’s potential to produce LH. Since estrogen levels are lower in the FF of EMS patients, the antagonistic actions of GnSAF against LH production are likely to result in suboptimal LH levels and impaired ovulation [55].

Various studies have explored changes in the composition of follicular fluid (FF) in women with EMS, including cytokines [56,57], oxidative stress markers [58,59,60,61,62,63], metals [64], growth factors, prostaglandins [65], and macrophage activation patterns [66]. The evidence of oxidative stress (OS) in the follicular microenvironment of these women [58,59,60,61,62,63] suggests that both their peritoneal fluid (PF) and FF may contain substances detrimental to oocyte competence acquisition. In this regard, studies evaluating the effect of FF from infertile women with EMS on the in vitro maturation of bovine oocytes have shown spindle and chromosomal damage [67], which could be prevented by adding antioxidants to the maturation medium, implying a pro-oxidant microenvironment in the ovarian follicles of these women [68]. These alterations may result from OS-induced damage to oocyte cell structures. Evidence also indicates higher levels of 8-Hydroxy-2-Deoxy-Guanosine (8OHdG) in the FF of infertile women with EMS, suggesting oxidative DNA damage in cumulus-oocyte complexes and a possible mechanism contributing to impaired oocyte quality in these patients [59].

#### 3.1.3. Dysregulated Immune Function, Hormonal Imbalance, and Oxygen Species in Peritoneal Fluid

Altered conditions within the peritoneal fluid (PF) environment of individuals with EMS typically lead to the growth, proliferation, and inflammation of ectopic endometrial tissue [69]. Changes in both humoral and cell-mediated immunity have been observed in the peritoneal environment of EMS patients [70].

A pronounced inflammatory response, accompanied by heightened reactive species and cytokines, can create an unfavorable pelvic environment, which is reflected in the composition of PF in these women [71,72,73,74]. Studies have indicated that changes in the PF composition of women with EMS, encompassing alterations in cellular and humoral mediators [75,76,77,78], including pro-inflammatory cytokines such as TNF-alpha, interleukin (IL)-1β, IL-6, IL-8, IL-10, IL-13, IL-17, IL-33, Monocyte Chemoattractant Protein (MCP)-1, Macrophage Migration Inhibitory Factor (MIF), and Regulated on Activation, Normal T Cell Expressed and Secreted (RANTES) [79,80,81,82,83,84,85], chemokines [86], angiogenic factors [84,87,88], and increased activated macrophages, T-lymphocytes, and NK cells [69], may lead to chronic inflammation, lesion proliferation, and local hormonal imbalances, which, in turn, can result in poor oocyte quality, impaired sperm motility, embryo toxicity, and reduced endometrial receptivity [89].

Increased levels of E2 in the PF of EMS patients have been shown to stimulate the cyclo-oxygenase-2 (COX-2) enzyme, which subsequently upregulates Prostaglandin E2 (PGE2) production. Prostaglandin E2 is the most potent stimulator of aromatase expression in endometriotic tissue [52], leading to elevated E2 production, further promoting its own proliferation and growth [7]. E2 and PGE2 then upregulate COX-2, initiating a positive feedback loop, resulting in a sustained endometriotic state [9]. Pregnancy-associated plasma protein (PAPP-A), produced by the endometrium, ovary, and placenta, exhibits protease activity towards insulin-like growth-factor-binding protein-4 (IGFBP-4), which normally suppresses follicular E2 production. The protease activity of PAPP-A reduces Insulin-like Growth-Factor-Binding Protein-4 (IGFBP-4) levels, leading to increased levels of free Insulin-like Growth Factors (IGF), which synergize with Follicle-Stimulating Hormone (FSH). IGF, in conjunction with LH, increases androstenedione and testosterone production, which are then aromatized under the influence of FSH into Estrone (E1) and E2, respectively [54,90]. Typically, this conversion would occur in the follicular aromatase. However, due to decreased aromatase activity in the follicles of EMS patients, the conversion takes place in the endometriotic tissue where there is an abnormally increased expression of aromatase, further accentuating the endometriotic state. This is supported by a study reporting increased PAPP-A levels in the peritoneal microenvironment of EMS patients, with the degree of elevation correlating with disease severity [91].

The production of abundant amounts of Reactive Oxygen Species (ROS) by elevated numbers of macrophages and polymorphonuclear leukocytes in the PF of EMS patients has been documented [92]. Two studies have also found increased ROS in EMS PF, although the levels were not significantly different from those in disease-free controls [93,94]. Based on these studies, it is believed that the development of oxidative stress (OS) in the local peritoneal environment may be one of the contributing factors in the chain of events leading to EMS-associated infertility [95]. It has also been proposed that redox levels may modulate the severity, dynamics, and progression of the disease. The increase in both the number and activity of macrophages in EMS is accompanied by the release of additional cytokines and other immune mediators, such as nitric oxide (NO) [96]. Nitric oxide is a free radical and a regulator of apoptosis [97]. Low levels of NO play an essential role in ovarian function and implantation. However, higher levels of NO and nitric oxide synthase (NOS) are observed in the endometrium of women with EMS [98].

In the PF of women with EMS-associated infertility, the total antioxidant capacity is reduced, and individual antioxidant enzymes, such as superoxide dismutase, are significantly lower [99]. Moreover, lipid peroxide levels are highest among patients with EMS, suggesting a potential role for ROS in the development of EMS [71]. An imbalance between antioxidants and oxidants has been reported in many studies investigating PF from women with EMS. It has also been theorized that ROS may contribute to the formation of adhesions associated with EMS. Although adhesions due to EMS are known to reduce fertility, the precise mechanism remains incompletely understood [100]. Additionally, alterations in folliculogenesis, likely caused by OS, may impair oocyte quality and have been proposed as a cause of subfertility associated with EMS. Levels of the OS marker, 8-hydroxy 1-deoxyguanosine, were higher in patients with EMS compared to those with tubal, male-factor, or idiopathic infertility [101].

OS has also been shown to induce genomic and mitochondrial DNA damage [68], directly resulting in reduced fertility [102]. It was observed that spermatozoa exhibited heightened DNA fragmentation when exposed to peritoneal fluid (PF) from patients with EMS. Moreover, the degree of fragmentation increased in correlation with the stage of EMS and the duration of infertility [103]. Similarly, oocytes displayed increased DNA damage when subjected to PF from EMS patients, with the extent of damage being influenced by the duration of exposure to the PF [104]. As anticipated, embryos cultivated in the PF of EMS patients also exhibited DNA fragmentation [105]. This elevated DNA damage in sperm, oocytes, and resulting embryos is believed to contribute to the higher rates of miscarriages, as well as fertilization and implantation failures, observed in individuals with EMS [103].

#### 3.1.4. Sperms Dysfunction

The presence of activated macrophages, cytokines, elevated TNF-alpha levels, heightened growth factors, and oxidative stress (OS) within the peritoneal fluid (PF) of infertile women affected by EMS can potentially exert toxic effects on sperm function [105,106,107,108]. These altered factors may lead to reduced sperm motility [107,109], induce sperm DNA fragmentation [108], promote abnormal sperm acrosome reactions [110], disrupt sperm membrane permeability or integrity [111], impair the interaction between sperm and the uterine tube epithelium [112], and hinder sperm–oocyte fusion [106]. These mechanisms represent additional potential contributors to infertility in individuals with EMS.

#### 3.1.5. Implantation Failure and Impaired Endometrial Receptivity

Uterine receptivity refers to the capacity of the endometrium to facilitate the normal implantation of an embryo [113]. When uterine receptivity is compromised, it can lead to various reproductive issues, ranging from the inability of embryos to implant (resulting in infertility) to incomplete or insufficient implantation (which may result in miscarriages). Histologically, uterine receptivity is established during the mid-luteal phase, commonly referred to as the “window of implantation” (WOI). This critical period is characterized by the occurrence of decidualization, a process that takes place in each menstrual cycle independently of the presence of an embryo [114,115]. However, conditions like EMS can disrupt this delicate process. EMS can interfere with uterine receptivity through various mechanisms, including the dysregulation of important signaling pathways and molecules in endometrial stromal cells, alterations in the expression of genes within the endometrium, changes in cell physiology, and the development of vascular abnormalities, among other factors [114].

In a well-functioning endometrium, the signaling of progesterone (P4) and estrogen is intricately synchronized, following a pattern dictated by the phase of the menstrual cycle. This coordination plays a vital role to maintain a regular menstrual cycle, facilitating successful embryo implantation, and fostering the progression of pregnancy [116]. During the proliferative phase, estrogen stimulates the proliferation of epithelial cells, whereas P4 counters the effects of estrogen, marking the onset of the secretory phase. This phase change prompts stromal cells to undergo decidualization. The disruption in the balance of these hormones—characterized by P4 resistance and an overwhelming influence of estrogen dominance—leads to EMS-associated infertility [116,117].

P4 serves as a critical regulatory hormone, preparing the uterus for implantation and sustaining pregnancy. Its effects hinge on the progesterone receptor (PR), with two subtypes (PR-A and PR-B) acting as ligand-activated transcription factors. Their absence, as seen in PR-specific knockout, leads to pregnancy failure due to unchecked estrogen-driven epithelial cell growth and compromised stromal decidualization [118]. Female mice lacking both PRA and PRB experience reproductive issues, underscoring the importance of epithelial PR expression for successful embryo implantation, the containment of estrogen-driven epithelial proliferation, and stromal decidualization. When the endometrium responds inadequately to P4, it is termed P4 resistance, notably seen in EMS as an inability to activate PR [119]. This receptor expression links to various transcriptional pathways. Chromatin-remodeling protein ARID1A, integral to steroid hormone signaling, potentially regulates PR as a target gene. Reduced ARID1A levels are associated with infertile EMS patients’ endometrium [120].

The P4-triggered Ihh-COUP-TFII signaling axis orchestrates essential epithelial–stromal interactions for embryo implantation and stromal decidualization. Indian Hedgehog (Ihh) ligands engage patched receptors (Ptch1 and Ptch2), heightening COUP-TFII expression in stromal cells. COUP-TFII, an orphan nuclear receptor, influences proper embryo implantation and decidualization. Ihh’s orchestration of uterine receptivity and decidualization, via pathways like Ihh-Ptch1-COUP-TFII-Hand2-Bmp2 and Wnt4 signaling, is hindered in EMS [121]. EMS-associated molecular signaling disruptions, such as KRAS activation and elevated SIRT1/BCL6, may impede Ihh signaling in the endometrium. This signaling axis significantly mediates PR, thus impaired Ihh signaling contributes to EMS-associated progesterone resistance [122].

Wnt signaling, particularly the Wnt/β–catenin pathway, significantly contributes to blastocyst activation, uterine development, and decidualization. During mouse uterine decidualization, Wnt signaling dynamically expresses diverse Wnt ligands, Frizzled Receptors (Fzd), inhibitors (Dkk1, Sfrps), and transcriptional activators (β-catenin, etc.). Bone morphogenetic protein 2 (Bmp2), the downstream of PR, acts as a key paracrine factor, transmitting embryonic adhesion signals from the epithelium to stroma, initiating decidualization. Bmp2’s role in decidualization through FK506-binding proteins (Fkbps) and Wnt ligands is evident in conditional knockout mouse models. Uterine β-catenin knockout impairs endometrial decidualization [123]. PR interacts with nuclear transcription factors to regulate decidual processes. FOXO1 collaborates with PR to promote the expression of decidua-related genes (IGFBP1, PRL). Reduced FOXO1 expression in EMS patients’ secretory endometrium is attributed to hyperactive PI3K/AKT pathways [124,125]. Mutations in PR lead to sterility in mice, characterized by diminished or absent ovulation, uterine hyperplasia, the absence of endometrial decidualization, and the restricted development of mammary glands [116].

Estrogen plays a pivotal role in regulating the transition of the endometrium into a receptive state. In a mice model of delayed implantation treated with P4, researchers observed that the receptive window’s duration was extended with lower estrogen levels, but rapidly shortened with higher concentrations [126]. Thus, the estrogen levels dictate the receptive window’s duration in the uterus. EMS is an estrogen-dependent gynecological disorder characterized by excessive estrogen signaling, leading to an elevated production of 17-β estradiol (E2). This affects both the eutopic endometrium and ectopic sites. Dysregulated 17β-HSD1 expression, rather than aromatase anomalies, has been linked to hyperestrogenism [127].

Maintaining an optimal balance between pro- and anti-inflammatory factors at the maternal–fetal interface is crucial for a successful pregnancy. Excessive inflammation is unfavorable for embryo implantation. Both peripheral blood (PB) and peritoneal fluid (PF) in EMS patients contain an array of substances secreted by endometriotic implants and immune cells, such as growth factors, steroid hormones, and inflammatory agents, creating an inflammatory microenvironment [128,129]. The profile of pro- and anti-inflammatory cytokines evolves dynamically in EMS patients. In EMS, the cytokine profile in PF shifts from unfavorable (e.g., IL-1β, IFN-γ, and TNF-alpha) to favorable (e.g., IL-4, IL-10, and TGFβ) for pregnancy as the condition progresses [130,131]. Elevated levels of TNF-alpha, IL-1, IL-6, and IL-17 in PF are generally observed in EMS patients [83]. Research by Lessey et al. revealed that PF negatively affects the endometrium (specifically LIF and integrin αvβ3) in EMS-afflicted women [132], suggesting that inflammatory cytokines in PF directly impact the endometrium. The study also found higher levels of pro-inflammatory cytokines (IL-1α, IL-1β, and IL-6) in the endometrial fluid of EMS patients [133].

The immune cells implicated in the initiation and progression of endometrial lesions encompass macrophages, neutrophils, NK cells, dendritic cells, and regulatory T cells (Treg). These immune cells release chemokines and cytokines, crucial for communication that influences endometrial receptivity for embryo implantation. Emerging evidence suggests the altered immune status in both endometrium and peritoneal environment in EMS patients, contributing to infertility and pregnancy failure [134,135]. Uterine NK (uNK) cells are crucial for successful embryo implantation and pregnancy. They contribute in embryo bio-sensing and impact endometrial fate decisions during implantation [136]. In cases of EMS, elevated CD16+ uNK cells might lead to an increased infertility risk due to an adverse inflammatory environment affecting implantation and decidualization [137].

Methylation changes in the Human Homeobox A10 (HOXA10) genes are significant as their dysregulation can contribute to EMS. HOXA10 expression is regulated by estrogen and progesterone [130]. These genes are pivotal for normal endometrial changes during the menstrual cycle, governing growth, differentiation, and embryo implantation [137,138]. In EMS patients, reduced HOXA10 expression during the secretory phase leads to decreased uterine receptivity and related infertility [136,138].

### 3.2. The Pathological Role of miRNAs in Endometriosis

Recent research has illuminated the presence of an abnormal array of miRNAs characterizing EMS, exerting a significant influence on the expression of relevant target mRNAs [139]. miRNAs encompass a diverse range, contributing to various stages of EMS. These molecules exhibit remarkable stability and can be found both intracellularly and in various body fluids [140,141]. Their altered expression in both blood and endometrial tissues suggests their potential relevance to the pathology of the EMS as well as EMS-associated infertility.

Several studies have reported changes in the expression of specific miRNAs in endometriotic lesions. These miRNAs include miR-1, miR-29c, miR-34c, miR-100, miR-141, miR-145, miR-183, miR-196b, miR-200a, miR-200b, miR-200c, miR-202, miR-365, and miR-375 [142]. Some of these miRNAs are known to play roles in processes such as Epithelial-Mesenchymal Transition (EMT), angiogenesis, cell proliferation, cell adhesion, and invasion [142,143]. EMT and angiogenesis are crucial components of the pathophysiology of endometriotic lesion formation. EMT is associated with cell migration and invasion during lesion development, while angiogenesis is required to establish a blood supply for the growing lesions.

In a related discussion by Nothnick, additional dysregulated miRNAs are highlighted, including miR-15, miR-20a, miR-23a/b, miR-29c, miR-126, miR-142, miR-145, miR-183, miR-199a, and miR-451. Distinctions are drawn between miRNAs that likely serve as key drivers of the disease, influencing cell proliferation, invasion, and angiogenesis [144].

Ohlsson Teague et al. analyzed paired samples of eutopic and ectopic endometrial tissue in EMS patients. Microarray analysis revealed the differential expression of 22 miRNAs, with 14 being upregulated (miR 145, miR 143, miR 99a, miR 99b, miR 126, miR 100, miR 125b, miR 150, miR 125a, miR 223, miR 194, miR 365, miR 29c, and miR 1) and 8 downregulated (miR 200a, miR 141, miR 200b, miR 142 3p, miR 424, miR 34c, miR 20a, and miR 196b). Among these, miR-145 was the most upregulated, while miR-141 was the most downregulated [145].

Cho et al. compared miRNA expression in serum samples of women with and without EMS. They found that miRNAs miR-135b, let-7b, let-7d, and let-7f were expressed at lower levels in EMS patients. They suggested that let-7b’s role in EMS might involve the dysregulation of the p53 pathway and cell cycle control [146].

Profiling peripheral blood in women with Stage III and IV EMS compared to those without EMS revealed the differential expression of 27 miRNAs. Six of these (miR-17-5p, miR-20a, miR-22, miR-15b-5p, miR-21, and miR-26a) were significantly downregulated in EMS. This study also observed altered levels of angiogenesis-related factors like VEGF A and TSP-1 [147,148]. miR-20a’s role in EMS remains controversial, with some studies indicating its upregulation in EMS [149]. Its function in angiogenic gene regulation requires further investigation to clarify its role in EMS pathogenesis.

Braza Boïls et al. found the lower expression of miR-449b-3p in ovarian endometriomas compared to eutopic endometrium. Additionally, diseased eutopic endometrial tissue showed lower levels of miRNAs miR-202-3p, miR-424-5p, miR-449b-3p, and miR-556-3p, along with higher levels of VEGF A. These findings suggest a potential role of miRNAs in regulating angiogenic activity in EMS [150]. Zolbin et al. conducted a study on adipocyte cells transfected with miRNA mimics and inhibitors (let-7b and miR-342-3p). They found that altered miRNA levels may alter the expression of genes involved in brown adipocyte differentiation, appetite, insulin sensitivity, and fat metabolism, potentially contributing to the low BMI phenotype observed in EMS patients [151].

Enhanced miR-146 levels marked EMS patients, with a greater expression in those suffering from pain symptoms and reduced Interferon Regulatory Factor 5 (IRF5) expression, which negatively regulates inflammation. This finding suggested an important role of the miR-146b level and variants in EMS [152]. Another individual study of miRNA related to EMS showed that a lower level of miR-126-5p combined with elevated BCAR3 expression facilitated the migration and invasion of EMS stromal cells [153]. Research showed that the miR-200 family, pivotal in EMT, which is crucial for endometriotic lesion development, showed reduced expression in both endometriomas and endometriotic lesions [154,155]. Moreover, miR-214-3p downregulation was tied to inhibiting EMS lesion fibrosis by targeting Connective tissue growth factor (CCN2) [156]. The downregulation of miRNA-34a-5p in endometrial stem cells led to the increased expression of VEGFA, which in turn promoted angiogenesis and contributed to the development and progression of EMS [157].

### 3.3. The Pathological Role of miRNA in Endometriosis-Associated Infertility

Extensive research has firmly established the adverse effects of EMS on fertility. Within this context, miRNA investigations have played a pivotal role (Table 1) in elucidating the intricacies of how EMS contributes to infertility.

Microarray assessments spotlighted diminished miR-543 expression in endometrial tissue from individuals grappling with EMS-associated infertility during the Window of Implantation (WOI), suggesting potential repercussions for embryo implantation [158]. Elevated miR135a and miR135b were identified in the endometrial tissue of EMS patients, highlighting how their overexpression downregulates implantation-related genes, including HOXA10 [159]. Increased miR-29c expression in the endometrial tissue of EMS patients correlated with impaired FK506 Protein Binding 4 (FKBP4) (Decidualization marker) expression, potentially contributing to progesterone resistance [160]. miR-194-3p plays a role in the development of progesterone resistance in EMS, leading to impaired fertility. This occurs through its suppression of PR levels and the decidualization process in the eutopic endometrium [161]. The expression of miR-2861 was found to be downregulated in endometriotic tissues. Decreased levels of miR-2861 might encourage the growth of ectopic endometrial cells in EMS by promoting cell proliferation and inhibiting apoptosis. Research has revealed a significant impairment in the regulation of apoptosis within the endometrium of infertile women with tubal factor and endometriosis. They have observed a significant inhibition of apoptosis in the endometrium of infertile women with EMS during the implantation window [162,163].

In the context of EMS, overexpressed miR-33b decreases Wnt/β-catenin signaling and inhibits the zinc finger E box-binding homeobox 1 (ZEB1) protein expression, which seemingly instigates EMS progression and impaired endometrial decidualization [164,165]. Moreover, miR-488 has been found to have inhibitory effects on certain aspects of cell behavior in EMS. Specifically, miR-488 targets a protein called Frizzled-7 (FZD7), which is a part of the Wnt signaling pathway. The Wnt pathway is important for cell proliferation, migration, and invasion also. In EMS, miR-488 overexpression reduced the levels of FZD7, which in turn downregulated the Wnt pathway. This led to a decrease in the proliferation, migration, and invasion of endometrial glandular epithelial cells in mice with EMS [166].

In the analysis of the ectopic endometrium, researchers observed a notable decrease in the expression of miR-141-3p compared to both the eutopic endometrium and endometrium samples from healthy controls. This particular miRNA appears to play a crucial role in the regulation of apoptotic processes. Its reduced expression led to an increase in BCL-2 levels while decreasing bax expression, thereby inhibiting apoptosis. This effect is achieved through the targeting of Krüppel-like factor 12 (KLF-12). Notably, the downregulation of miR-141-3p had a significant impact on elevating the protein levels of KLF-12 [167,168,169,170]. Moreover, several studies have indicated that KLF-12 functions as a negative regulator during the decidualization of endometrial stromal cells (ESCs) [171,172,173]. Downregulated miR-205-5p reduced apoptosis and promoted migration and invasion in the ectopic endometrium by targeting the angiopoietin-2 (ANGPT2) [163,174].

The downregulation of miR-138 expression induced inflammation through the NF-κB signaling pathway and reduced apoptosis via the NF-κB/VEGF signaling pathway in EMS. Some studies have investigated the implications of immune dysfunction and the inhibition of apoptosis in endometriosis-associated infertility, highlighting how immune system dysfunction and the reduction of apoptosis can affect fertility in individuals with EMS [6,89,128,163,175]. The increased expression of miR-370-3p induced apoptosis and suppressed cell proliferation in endometriotic cells. Simultaneously, Steroidogenic factor 1 (SF-1) inhibited the decidualization process in the endometrium, leading to infertility. This also led to the abnormal development of uterine glands, shifts in immune homeostasis, and physiological inflammatory responses [176,177]. Among EMS patients, the miR-34 family exhibited downregulated expression across three members, implying that miR-34a reduced apoptosis and miR-34c, and miR-34b downregulation potentially governed progesterone resistance, promoting proliferation and ectopic tissue survival [171,172]. In the eutopic endometrium of patients with EMS, overexpressed miR-196a upregulated MEK/ERK signaling through PR downregulation and inhibited decidualization [173].

Downregulated miR-9 has been identified as a regulator targeting BCL-2, an anti-apoptotic protein that exhibits increased expression in females diagnosed with EMS [163,178]. miR-451 showed downregulation in the eutopic endometrium with EMS and YWHAZ, OSR1, TTN, and CDKN2D were the potential target genes of the miRNA. This downregulation expression of the miRNA could be responsible for the proliferation promotion and apoptosis inhibition [163,179]. Progesterone has been demonstrated to exert regulatory control over miR-125b. By modulating the expression of one of its target genes, MMP26, this regulatory mechanism can influence endometrial receptivity in women undergoing in vitro fertilization procedures [180]. Upregulated miR-139-5p in ectopic stromal cells with EMS suppressed the impact of the HOX genes HOXA9 and HOXA10 [181]. A study showed that the endometrium expressed both HOXA9 and HOXA10 genes at elevated levels, and these genes had crucial functions in regulating endometrial receptivity [182].

miR-210-3p has been implicated in the pathogenesis of EMS, with research indicating its upregulation in both the eutopic and ectopic endometrial tissues of women diagnosed with EMS when compared to the endometrial tissues of healthy women. This specific miRNA is believed to play a role in processes such as cell proliferation and the response to DNA damage caused by oxidative stress through targeting the BARD1 gene [102,183]. miR22-5p was found to be downregulated in the eutopic endometrium of women with EMS during the secretory phase. During the implantation window in the human endometrium, there is a notable upregulation of ten-eleven translocation (TET2), a key marker involved in DNA hydroxy-methylation. This upregulation of TET2 and estrogen receptor 2 (ESR2) expression are directly controlled by miR22-5p [184]. Increasing the expression of miR-27b-3p led to the inhibition of HOXA10 expression, ultimately leading to increased proliferation, migration, and invasion capabilities in EMS cells [185,186].

In progesterone-resistant EMS, there was an observed increase in the expression of miR-92a. This higher miR-92a expression was found to be inversely associated with reduced PTEN (Phosphate and Tension Homolog Deleted on Chromosome 10) expression. The elevated levels of miR-92a were linked to enhanced cell proliferation and the development of progesterone resistance in EMS patients [187]. A reduction in the levels of long noncoding RNA H19, leading to its binding with miRNA let-7, had resulted in increased let-7 activity within the eutopic endometrium in EMS patients. Research had demonstrated the significant roles played by H19 and let-7 in influencing the IGF signaling pathway. As a consequence, the diminished IGF1 signaling pathway had hindered stromal cell proliferation in the eutopic endometrium, ultimately leading to impaired endometrial receptivity [188]. Figure 2 illustrates the pivotal role of miRNA in EMS-associated infertility.

**Table 1 biomedicines-11-03087-t001:** Role of Dysregulated Key miRNAs in Endometriosis-associated Infertility.

miRNAs	Dysregulation	Target Gene/Signaling Pathway	Effects	Ref.
miR-543	Downregulated	-	Affect embryo implantation	[158]
miR-135a/b	Upregulated	HOXA10	Downregulate the expression of implantation-related genes, including HOXA10	[159]
miR-29c	Upregulated	FKBP4	Lead to impaired expression of FKBP4 (Decidualization marker)—progesterone resistance	[160]
miR-194-3p	Upregulated	PR	Progesterone resistance, inhibit decidualization	[161]
miR-2861	Downregulated	STAT3, MMP2	Upregulate STAT3 and MMP2 expression and thus promote proliferation and inhibit apoptosis of ectopic endometrial cells in EMS	[162,163]
miR-33b	Upregulated	Wnt/β-catenin	Decrease Wnt/β-catenin signaling, inhibit the zinc finger E-box-binding homeobox 1 (ZEB1) protein expression, impair endometrial decidualization	[123,164,165]
miR-488	Downregulated	Wnt signaling	Suppress Wnt signaling, inhibit endometrial glandular cell proliferation, migration, and invasion	[123,166]
miR-141-3p	Downregulated	KLF12	Inhibit apoptosis, induce cell proliferation and migration decrease decidualization in ectopic endometrial stromal cells (ESCs)	[167,168,169,170]
miR-205-5p	Downregulated	ANGPT2	Reduce apoptosis and promote migration and invasion in ectopic endometrium	[163,174]
miR-138	Downregulated	NF-κB/VEGF	Induce inflammation, reduce apoptosis via inhibition of nuclear factor-NF-κB and VEGF signaling pathway	[175]
miR-370-3p	Downregulated	SF-1	Reduce apoptosis, induce cell proliferation, inhibit decidualization—results in infertility	[176,177]
miR-34a/b/c	Downregulated	SIRT1	Progesterone resistance, enhance proliferation and ectopic tissue survival	[171,172]
miR-196a	Upregulated	MEK/ERK	Progesterone resistance, decrease decidualization	[173]
miR-9	Downregulated	BCL-2	Reduce apoptosis	[163,178]
miR-451	Downregulated	YWHAZ, OSR1, TTN, and CDKN2D	Promote proliferation and inhibit apoptosis	[163,179]
miR-125b	Upregulated	MMP26	Change progesterone level and influence endometrial receptivity	[180]
miR-139-5p	Up-regulated	HOXA9, HOXA10	Impair endometrial receptivity	[181,182]
miR-210-3p	Up-regulated	BARD1	Promote cell proliferation, response to DNA damage caused by oxidative stress	[102,183]
miR22-5p	Downregulated	TET2	Regulate estrogen receptor 2 expression, DNA methylation	[184]
miR-27b-3p	Upregulated	HOXA10	Enhance cell proliferation, migration, and invasion	[185,186]
miR-92a	Upregulated	PTEN	Progesterone resistance, higher cell proliferation	[187]
miRNA Let-7 family	Upregulated	H19 IncRNA, IGF1R, KRAS	Inhibit eutopic endometrial cell proliferation, leading to impaired endometrial receptivity	[188]

## 4. A Possibility of Using miRNA as a Diagnostic Marker for Endometriosis

miRNAs have highly stable structures and play a significant role in regulating various pathological processes during the formation of EMS, they are involved in controlling the adhesion, invasion, apoptosis, angiogenesis, and proliferation of endometriotic cells [185,189,190]. Within the cells, miRNAs directly participate in the post-transcriptional regulation of target genes, while outside the cells, they can bind to proteins or extracellular vesicles, thus avoiding degradation by ribonucleases [165,191]. This allows them to remain stable in the bloodstream, suggesting that miRNAs have the potential to serve as ideal diagnostic markers. Over the past decade, the application of miRNA in the diagnosis of EMS is one of the hot topics in the field of miRNA research. Numerous studies have reported various miRNA types that could be utilized for diagnosing EMS. Among these, miR-451, the miR-200 family, miR-199, miR-125, and let-7-family have consistently demonstrated good diagnostic performance in multiple studies, making them promising candidate biomarkers for diagnosing EMS [146,192,193,194,195].

### 4.1. miR-451

Multiple studies have demonstrated that the expression levels of miR-451 are upregulated in both the blood and tissue of patients with EMS. In the year 2020, a prospective clinical study was conducted by Sarah and colleagues [196]; the study included 100 patients who required gynecological laparoscopic surgery for various reasons. These patients were divided into two groups, the EMS group and the control group, based on whether endometriotic lesions were found during laparoscopy. Prior to the surgery, peripheral blood samples were collected from these 100 patients, serum was obtained from the peripheral blood through centrifugation, and real-time quantitative PCR was employed to examine the levels of miRNAs in the subjects’ serum. The results revealed significant increases in the levels of miR-451a, miR-125b-5p, miR-150-5p, and miR-324-3p in the EMS group. Conversely, the levels of miR-3613-5p and let-7b were significantly decreased in this group. When miR-451a was used as a standalone diagnostic marker for EMS, the area under the curve (AUC) was 0.84, and the sensitivity and specificity were 90.0% and 91.2%, respectively. These findings suggest that miR-451a could serve as a promising biomarker for predicting EMS. In another two previous studies, the Receiver Operating Characteristic (ROC) curves for miR-451a were 0.835 and 0.86 [197,198], demonstrating the favorable reproducibility of miR-451a as a biomarker. Furthermore, this study included patients who underwent treatment with estrogen and progestin hormones, in contrast to many previous studies which had excluded patients receiving hormonal treatment, and the results indicated that there was no significant difference in the serum miRNA levels based on whether hormonal treatment was administered. This further broadens the scope of using miRNAs as biomarkers for diagnosing EMS across different patient populations.

Reportedly, 30% to 50% of patients with EMS also experience infertility. In these types of patients, the levels of miR-451a in the serum are also significantly elevated [199]. This suggests that miR-451a can serve as a potential biomarker for predicting both EMS-associated infertility patients.

### 4.2. miR-199

Research has confirmed a significant elevation in the levels of miR-199a in the blood of patients with EMS. In 2017, Ahmed M and colleagues conducted a prospective cohort study in Egypt, which for the first time incorporated a substantial sample size (45 cases in the group and 35 cases in the control group) for investigation [200]. The results demonstrated a marked increase in miR-199a levels in both the serum and peritoneal fluid of the EMS group, and sensitivity and specificity for diagnosing EMS were both 100%, with an AUC value of 1 in the ROC curve, indicating that miR-199a could serve as a reliable indicator for diagnosing EMS. These findings align with the value of miR-199a in the EMS diagnosis, as first published by Wang et al., in 2013 [201]. Wang’s research indicated a correlation between elevated miR-199a levels and the severity of lesions. However, this correlation was not confirmed in Ahmed M’s study, where the authors speculated that this discrepancy might be attributed to differences in age, ethnicity, and sample sources among the participants.

### 4.3. miR-125

The expression of miRNA-125b is significantly upregulated in the serum of patients with EMS. In a clinical study conducted in 2016 by Cosar E and colleagues [198], which included 48 study subjects, 24 of whom were patients with EMS (the experimental group) and 24 were patients undergoing laparoscopic surgery for various benign gynecological conditions (the control group). The levels of ten miRNAs in the serum of both groups were analyzed and the results showed that in the serum of patients with EMS, miRNA-125b-5p, miRNA-150-5p, miRNA-342-3p, miRNA-143-3p, miRNA-145-5p, miRNA-500a-3p, miRNA-451a, and miRNA-18a-5p were upregulated by more than 10-fold, while the miRNA-3613-5p and miRNA-6755-3p levels were significantly decreased. Among these, miRNA-125b-5p had the highest AUC value (0.97). When miRNA-125b-5p, miRNA-451a, and miRNA-3613-5p were combined for diagnostic purposes, they achieved a sensitivity of 100% and a specificity of 96%, with an AUC of 1, indicating an excellent diagnostic value. Subsequent research also confirmed the aberrant expression of miRNA-125b-5p in the serum of patients with EMS. Abdel-Rasheed M and colleagues compared the expression profiles of miRNAs in the serum of patients with severe EMS and normal females [202]. They identified 32 differentially expressed miRNAs, among which miRNA-125b was upregulated by 12-fold, suggesting that miRNA-125b could serve as a novel marker for diagnosing EMS.

### 4.4. Let-7 Family

Let-7b is one of the earliest studied miRNA groups and plays a role in gene regulation within cells. By binding to the mRNA of target genes, it inhibits the translation or degradation of those genes, participating in biological processes such as cell cycle regulation, cell differentiation, cell migration, and growth [203]. Existing research has confirmed its association with various malignant tumors, kidney diseases, brain disorders, and reproductive system disorders [204,205,206,207]. Due to Let-7’s negative regulation of the Ras oncogene family, it is classified as a tumor suppressor, and the downregulation of the Let-7 family has been observed in many malignant tumors.

The pathological processes of EMS involve abnormal adhesion, invasion, angiogenesis, proliferation, and apoptosis of endometrial cells in other tissues, exhibiting biological behaviors similar to malignant tumors [208]. In many studies, the abnormal expression of the Let-7 family has been found in the peripheral blood serum of patients with EMS. Narges and colleagues discovered that in the peripheral blood serum of EMS patients [193], the levels of Let-7d-3p and miR-224-5p were significantly downregulated, while the level of miR-199b-3p was significantly elevated. When used individually for diagnosing EMS, their AUC values were 0.694, 0.914, and 0.843, respectively. However, when these three were combined for diagnosing EMS, the AUC increased to 0.992 (sensitivity 96%, specificity 100%), indicating that a combined diagnostic approach can enhance the accuracy of early EMS diagnosis.

EMS is a hormonally dependent condition and, theoretically, the levels of miRNAs in the peripheral blood of patients should vary with the menstrual cycle. Cho S and colleagues [193] showed that Let-7b/c/d/e expression significantly decreased in the proliferative phase in patients with EMS based on a stratified study of the menstrual cycle.

### 4.5. miR-200 Family

The miR-200 family consists of miR-200a/b/c, miR-141, and miR-429. The initial research indicated that members of the miR-200 family play a crucial role in regulating an epithelial–mesenchymal transition (EMT). EMT involves the transformation of cells from an epithelial type to a mesenchymal type, enabling cancer cells to acquire invasive and metastatic capabilities [209]. EMS involves pathological physiological processes such as abnormal adhesion, invasion, proliferation, and apoptosis of endometrial cells, sharing biological behaviors similar to cancer cells. Therefore, it has been suggested that the miR-200 family participates in the development of EMS by mediating the EMT process. Compared to in situ endometrial tissue, the expression of miR-200a/b and miR-141 is downregulated in ectopic endometrial tissue [210].

A more recent study has investigated miR-202 in exosomes from vaginal discharge as a possible indicator of EMS. Exosomes are small vesicles released by cells and can contain biomolecules, such as microRNAs, which can be used as diagnostic markers for various conditions. Further research is likely required to validate the efficacy of this potential biomarker [211].

In a study conducted in 2015 by Rekker K et.al [212], it was found that the levels of miR-200a/b and miR-141 in the peripheral blood of EMS patients were significantly reduced, with miR-200a showing the most pronounced decrease in stage III-IV EMS patients, demonstrating superior diagnostic potential. Furthermore, this study also discovered variations in the levels of miR-200a/b and miR-141 at different times of the day. In patients with stage I-II EMS, the peripheral blood levels of miR-200a/b and miR-141 significantly decreased at night. Therefore, the authors suggested considering the timing of sample collection when using miRNAs as diagnostic markers.

One of the most significant harms of EMS to women is infertility, and the mechanisms underlying infertility associated with EMS are a current research focus. With the important role of miRNAs in the occurrence and development of EMS being discovered, many miRNAs that can be used for diagnosing infertility related to EMS have gradually emerged. In 2017, Xu and colleagues conducted a comparative analysis of the uterine endometrium in the luteal phase of eight EMS-related infertility patients and six other infertility patients using high-throughput sequencing [213]. They identified 57 newly dysregulated miRNAs and 61 known miRNAs, among which three upregulated miRNAs (hsa-miR-1304-3P, hsa-miR-5446, and hsa-miR-3684) were not expressed in the control group, and two downregulated miRNAs (hsa-miR-3935 and hsa-miR-4427) were undetectable in the patient group. This suggests that these five miRNAs may have a stronger relationship with infertility related to EMS compared to other miRNAs, warranting further investigation into their role in the diagnosis of EMS-related infertility.

A recent study analyzed miRNAs in saliva samples from 153 EMS patients (36 infertile and 117 fertile) and identified 34 dysregulated miRNAs [214]. Their sensitivity for diagnosing infertility ranged from 0% to 97.2%, specificity from 1.7% to 100%, and AUC from 34.6% to 65.4%. Among these, miR-6818-5p, miR-498, miR-1910-3p, miR-3119, and miR-501-5p showed a higher diagnostic value. For the first time, this study analyzed differences in miRNA expression in saliva samples from patients with EMS-associated infertility, which could aid in the early detection of EMS-associated infertility and facilitate timely treatment.

In summary, current research has reported numerous miRNAs (Table 2) that can be used for the diagnosis of EMS and EMS-related infertility. When summarizing the results of various studies, miR-451, the miR-200 family, miR-199, miR-125, and let-7a-f have shown good repeatability and diagnostic efficiency, potentially serving as candidate biomarkers for the diagnosis of EMS [147,215,216,217,218]. However, due to differences in clinical sample selection and experimental methods, there is still no widely recognized marker for diagnosing EMS. Research on the use of miRNA for the diagnosis of EMS-related infertility is limited, and larger prospective studies are needed to identify specific miRNAs that can be used for the early diagnosis of EMS-related infertility.

## 5. Therapeutic Perspective of miRNA in Endometriosis

Research has shown that certain miRNAs are associated with the pathogenesis of EMS, such as their involvement in regulating cell proliferation, inflammatory responses, and angiogenesis. The abnormal expression of these miRNAs is related to the development of EMS. Researchers are exploring the use of miRNAs as a novel approach for treating EMS, wherein experiments involve modulating the activity of specific miRNAs using miRNA mimics or miRNA inhibitors.

Previous studies have confirmed the downregulation of miRNA-200 expression in ectopic endometrial tissue. In vitro cell experiments conducted by Liang Z and colleagues demonstrated that the exogenous overexpression of miRNA-200c could inhibit the proliferation and expansion of primary endometrial stromal cells [219]. This process was achieved through the regulation of metastasis-associated lung adenocarcinoma transcript 1 (MALAT1). Furthermore, additional animal experiments showed that rats injected with miRNA-200c mimics exhibited a significant reduction in the volume of endometriotic lesions compared to rats receiving miRNA-200c inhibitors. This suggests that miRNA-200c mimics could serve as a novel therapeutic approach for EMS.

Some progress has been made in using miRNAs to address EMS-associated infertility. Research has revealed the relationship between miRNA expression levels and infertility in patients with EMS [145,220]. There have been attempts to use miRNA mimics or miRNA inhibitors to modulate the expression of these miRNAs, thereby improving the reproductive capabilities of patients with EMS-related infertility [221]. EMS is commonly associated with inflammation or immune system abnormalities, and certain miRNAs have been found to play a crucial role in inflammation and immune regulation. By modulating inflammatory responses and immune cell activity, miRNAs contribute to alleviating fertility issues caused by EMS [2]. Another issue often associated with EMS is the formation of new blood vessels. miRNAs can participate in the regulation of blood vessels by modulating genes related to angiogenesis or maintenance, inhibiting the formation of new blood vessels, and thereby aiming to treat EMS-related infertility [222].

Nayak R and colleagues retrieved four microarray datasets from the GEO databases [223], focusing on gene expression profiles in EMS and recurrent implantation failure. They screened for differentially expressed miRNAs (DE-miRNAs) in these databases, followed by an analysis of differentially expressed genes (DEGs) to identify key miRNAs that could be targeted to discover drugs capable of altering their expression. A total of 22 DE-miRNAs (11 upregulated and 11 downregulated) were identified in both EMS and infertile patients, these DE-miRNAs are considered to be highly influential in regulating EMS-related infertility. The expression of DE-miRNAs can be modulated by various types of upstream regulators. Four transcription regulators (CCND1, FOXO1, CREB1, and HDAC4) were found to be targeted by DE-miRNAs, suggesting that they could be critical regulators in events associated with EMS and may play a role in infertility. In the end, drugs were selected that could regulate these target transcription regulators (TRs). The results indicated that BRD-K17953061 maximally downregulated FOXO1 and CGP-60474 maximally upregulated CREB1. This suggests that these two drugs have the potential to reverse the effects of these two TRs implicated in EMS-associated infertility.

## 6. Conclusions and Future Direction

Alterations in the expression levels of numerous miRNAs have been observed in both endometriotic lesions and the eutopic endometrium of women with EMS compared to those without the condition [12,224]. These miRNAs appear to play a contributory role in the pathogenesis of EMS, influencing processes such as cell proliferation, apoptosis, immune responses, and inflammatory pathways [69,162,163]. While the association of miRNAs with EMS is undeniable, it is important to acknowledge that these associations are primarily based on observational findings rather than concrete proof-of-concept research. Currently, due to the absence of definitive proof-of-concept animal experiments, it remains unclear whether the alterations in miRNA expression in EMS are a consequence of the disease’s effects or if they actively participate in the underlying etiopathogenic mechanisms of EMS. To address this critical question, there is an urgent need for comprehensive proof-of-concept animal experiments within the field of EMS research. These experiments would help elucidate the precise role of miRNAs in the development and progression of EMS, shedding light on their true significance in the disease. Furthermore, the insights gained from such research could pave the way for the development of therapeutic agents based on miRNAs, offering new avenues for the treatment of EMS in the future.

While the precise etiopathogenetic role of miRNAs in EMS remains to be fully elucidated, the current findings regarding alterations in miRNA expression offer a promising opportunity for the development of miRNAs as diagnostic markers in EMS. miRNAs, once released from cells, circulate in stable forms in the bloodstream, protected from degradation [225]. They are encapsulated in extracellular vesicles, such as exosomes, or bound to proteins before being released into the extracellular space, ensuring their remarkable stability in various body fluids, including blood, urine, saliva, and cerebrospinal fluid [226]. Given this stability, circulating miRNAs can serve as valuable indicators of the pathological state of the endometrial tissue from which they originate. This stability factor facilitates the easy and non-invasive collection and analysis of miRNAs, rendering them highly attractive candidates for diagnostic markers. Consequently, it is now opportune to conduct large-scale statistical analyses correlating the clinical progression of EMS with the expression levels of miRNAs, paving the way for the development of miRNAs as diagnostic markers for this condition.

## Figures and Tables

**Figure 1 biomedicines-11-03087-f001:**
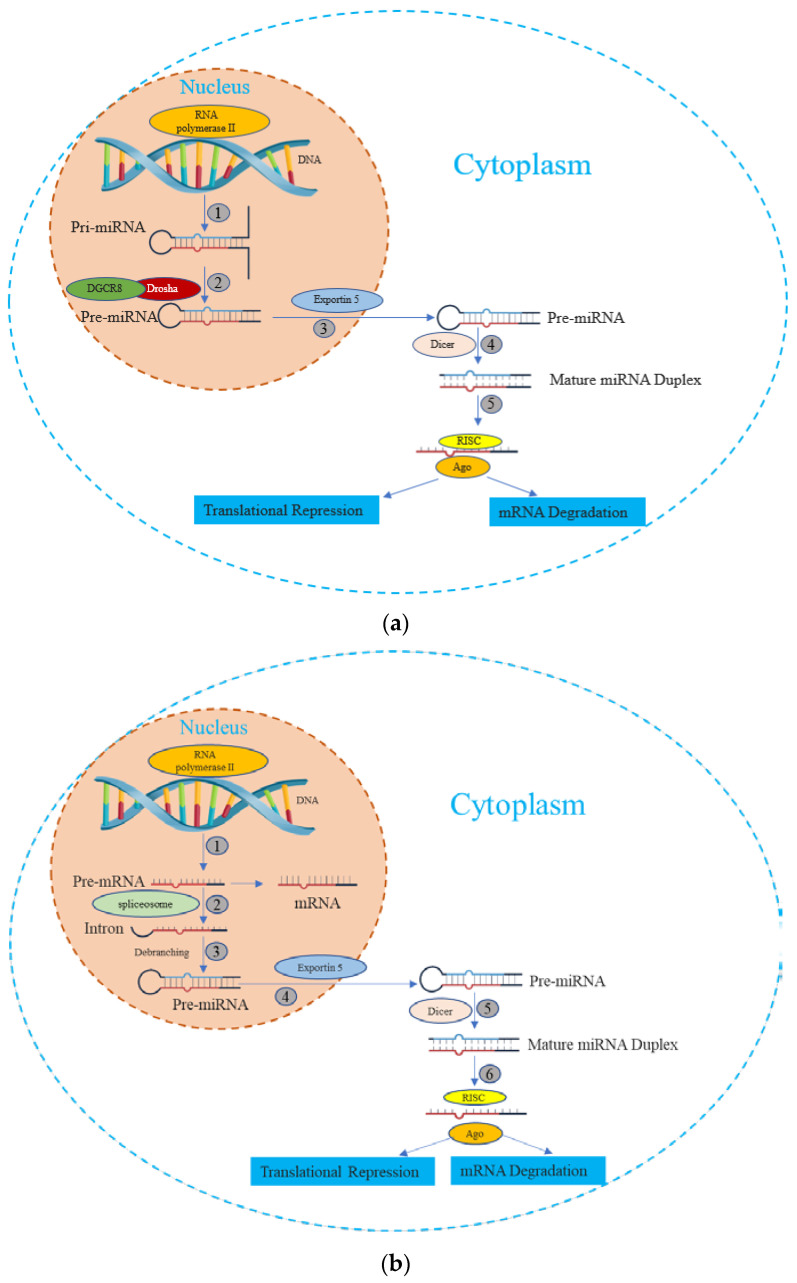
The miRNA biogenesis pathways. (**a**) Canonical Pathway of miRNA biogenesis: (1) The miRNA precursor is transcribed by RNA Polymerase II, giving rise to primary miRNA (pri-miRNA). (2) The pri-miRNA undergoes maturation processing involving nuclear cleavage by the Drosha enzyme, in conjunction with the cofactor DGCR8. This step results in the formation of pre-miRNA. (3) The pre-miRNA is transported from the nucleus to the cytoplasm via Exportin-5. (4) In the cytoplasm, the pre-miRNA undergoes additional maturation as Dicer cleaves one end of the miRNA, leading to the creation of a mature miRNA duplex. (5) Helicase action converts the mature miRNA duplex into single-stranded transcripts. The single-stranded miRNA transcript binds with a ribonucleoprotein complex called the RNA-induced silencing effector complex (RISC). This complex guides the miRNA to complementary mRNA sequences, resulting in diverse outcomes such as mRNA degradation, translational repression, or inhibition. (**b**) Non-Canonical Pathway of miRNA biogenesis: (1) The pre-mRNA is transcribed by RNA Polymerase II. (2) Spliceosome, a large RNA-protein complex removes introns from a transcribed pre-mRNA. (3) There is no involvement of Drosha activity; Pre-miRNAs are produced directly through the debranching of introns. (4), (5), and (6) Once pre-miRNAs are transported, they undergo processing similar to what occurs in the canonical pathway.

**Figure 2 biomedicines-11-03087-f002:**
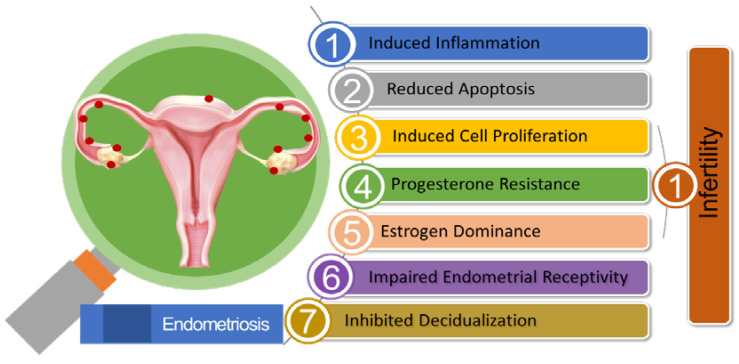
Summary of the Pathogenic Role of miRNA in Endometriosis-associated Infertility.

**Table 2 biomedicines-11-03087-t002:** miRNA Biomarkers for the Diagnosis of Endometriosis and Endometriosis-associated Infertility.

miRNAs	Dysregulation	EMS-Associated Infertility	Sensitivity (%)	Specificity (%)	AUC	Ref.
miR-106b-3p	Upregulated	Y	N/A	N/A	N/A	[215]
miR-451a	Upregulated					
miR-486-5p	Upregulated					
miR-1304-3p	Upregulated	Y	N/A	N/A	N/A	[213]
miR-544b	Upregulated					
miR-3684	Upregulated					
miR-3935	Downregulated					
miR-29c	Downregulated	Y	N/A	N/A	N/A	[216]
miR-200a	Upregulated					
miR-145	Upregulated					
miR-31	Downregulated	N/A	N/A	N/A	N/A	[217]
MiR-145	Upregulated					
miR-199a	Upregulated	N/A	78.33	76.00	0.825	[201]
miR-122	Upregulated		80.00	76.00	0.835	
miR-145	Upregulated		70.00	96.00	0.883	
miR-542-3p	Downregulated		79.66	92.00	0.854	
miR-17-5p	Downregulated	Y	combine	combine	combine	[195]
miR-20a-5p	Downregulated		of the	of the	of the	
miR-199a-3p	Downregulated		five	five	five	
miR-143-3p	Downregulated		0.96	0.79	0.93	
Let-7b-5p	Downregulated					
miR-125-5p	Upregulated	N/A	100.00	96.00	0.97	[198]
miR-451a	Upregulated		N/A	N/A	0.84	
miR-3613-3p	Downregulated		N/A	N/A	0.86	
miR-17-5p	Downregulated	N/A	70.00	70.00	0.74	[147]
miR-20a	Downregulated		60.00	90.00	0.79	
miR-22	Downregulated		90.00	80.00	0.85	
miR-17	Downregulated	N/A	N/A	N/A	0.84	[219]
miR-200a	Downregulated	N/A	90.60	62.5	0.75	[212]
miR-200b	Downregulated		90.60	48.5	0.67	
miR-141	Downregulated		71.90	70.8	0.71	
miR-199a	Upregulated	N/A	91.40	100.00	1.00	[201]
miR-122	Upregulated		95.60	100.00	0.96	
miR-125b-5p	Upregulated	N/A	N/A	N/A	0.97	[196]
miR-451a	Upregulated		N/A	N/A	0.84	
miR-3613-5p	Downregulated		N/A	N/A	0.76	

Note: N/A indicates data not available, Y indicates yes.

## Data Availability

Not applicable.

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
