# Peer review of "The Pathological Role of miRNAs in Endometriosis"

_biomedicines, 2023, doi:10.3390/biomedicines11113087_

Round 1
Reviewer 1 Report
Comments and Suggestions for Authors
Reviewer comments and suggestions
The authors of this study discussed the advancements made in understanding the pathological role of miRNA in endometriosis (EMS) and its association with EMS-associated infertility. Moreover, the review may contribute to the ongoing pursuit of developing miRNA-based therapeutics and diagnostic markers for EMS.
Overall, the manuscript needs to be modified. I have listed the concerns/comments that needed to be explained/modified.
- Line 32 (reference 2) Can you briefly explain the points for the common reader of your manuscript?
- Line 33-34 seems that grammatically incorrect line, please modify it
- Line 36 what was the percentage of this rate with not surgical treatment linking with fertile women?
- Line 38 as such you have written two times, please delete one
- Line 92-94 reference
- Figure 1 shows there was no figure for non-canonical pathways.
- Line 124 reference 33 needs to be explained.
- Line 131 (Numerous investigations have been undertaken to shed light on) but the authors did not mention a single reference for this line
- Line 157-158 is there any therapeutic study there to show that inhibiting tnf alpha may have reduced infertility or associated mechanism
- In line 194 a typo error was seen, please check and in line 209 there should be bracket.
- Line 229-230, please add suitable references for these lines
- Line 309-311 Suddenly the paragraph related to estrogen comes
- Line 327 Please check the expression of NK cells in the previous s paragraph.. as all should be consistent
- Section 404-407 was related to infertility however, infertility was not discussed and similarly, 404-409 no discussion of infertility only discussed wnt. which were already discussed in earlier sections.
- Line 427-430 Here also I could not find. The paragraph needs to be shortened and specific points on infertility should be present otherwise no issue of discussing the topic in another section
- Comment for table 2 table should be well prepared.
- Lines 619-620 need to explain more.
- General comments for reference list “Only 5 references were found for 2023, I think they should have to work out on the references part as well.
Author Response
Point-by-point response
Reviewer #1:
General Comment:
The authors of this study discussed the advancements made in understanding the pathological role of miRNA in endometriosis (EMS) and its association with EMS-associated infertility. Moreover, the review may contribute to the ongoing pursuit of developing miRNA-based therapeutics and diagnostic markers for EMS. Overall, the manuscript needs to be modified. I have listed the concerns/comments that needed to be explained/modified.
Response to General Comment:
Thank you very much for your excellent and through review of our manuscript, as well as for recognizing the significance of our review paper. Your comments have made our manuscript much better. We truly appreciate your time and effort. Thank you.
Comment 1.
Line 32 (reference 2) Can you briefly explain the points for the common reader of your manuscript?
Response to Comment 1:
It seems to be that our writing did not effectively convey our intended meaning. Based on your comment, we have rewritten the portion to provide a clearer description. Please refer to lines 28-34. Thank you.
Comment 2.
Line 33-34 seems that grammatically incorrect line, please modify it.
Response to Comment 2:
We rewrote the sentence. Please refer to lines 34-35. Thank you.
Comment 3.
Line 36 what was the percentage of this rate with not surgical treatment linking with fertile women?
Response to Comment 3:
Surgical treatment with laparoscopy in women with minimal or mild EMS increases the pregnancy rate to 30.7% from 17.7% in the non-treated group over a 20-week observational period. Reflecting your question, we rewrote the fertility issue associated with EMS. Please refer to lines 35-42. Thank you.
Comment 4.
Line 38 as such you have written two times, please delete one.
Response to Comment 4:
We deleted the redundancy. Please refer to line 42. Thank you.
Comment 5.
Line 92-94 reference.
Response to Comment 5:
We added references to the sentences. Please refer to lines105 and 107. Thank you.
Comment 6.
Figure 1 shows there was no figure for non-canonical pathways.
Response to Comment 6:
We have included the figure for the non-canonical pathway as per your suggestion. Please revised Figure 1. Your insightful feedback has greatly improved our paper. We truly appreciate your contribution. Thank you.
Comment 7.
Line 124 reference 33 needs to be explained.
Response to Comment 7:
According to your comment, we explained more precisely. Please refer to lines 149-151. Thank you.
Comment 8.
Line 131 (Numerous investigations have been undertaken to shed light on) but the authors did not mention a single reference for this line.
Response to Comment 8:
We added multiple references to the sentences to match the description. Please refer to line 159. Thank you.
Comment 9.
Line 157-158 is there any therapeutic study there to show that inhibiting tnf alpha may have reduced infertility or associated mechanism.
Response to Comment 9:
According to your comment, we added the case study results of the therapeutic application of TNF-alpha inhibitors in EMS. Please refer to lines 191-195. Thank you.
Comment 10.
In line 194 a typo error was seen, please check and in line 209 there should be bracket.
Response to Comment 10:
We corrected the mistakes. Please refer to lines 240 and 259. Thank you.
Comment 11.
Line 229-230, please add suitable references for these lines.
Response to Comment 11:
We added a reference to the sentence. Please refer to line 284. Thank you.
Comment 12.
Line 309-311 Suddenly the paragraph related to estrogen comes.
Response to Comment 12:
We discussed a possible association of estrogen with the pathogenesis of EMS. According to your comment, we modified our writing to clarify it. Please refer to lines 375-383. Thank you.
Comment 13.
Line 327 Please check the expression of NK cells in the previous s paragraph. as all should be consistent.
Response to Comment 13:
We corrected our inconsistency. Please refer to lines 179 and 246. Thank you.
Comment 14.
Section 404-407 was related to infertility however, infertility was not discussed and similarly, 404-409 no discussion of infertility only discussed wnt. which were already discussed in earlier sections.
Response to Comment 14:
Based on your comment, we have rewritten the portion to provide a clearer description. Please refer to lines 489-495 and 496-498. Thank you.
Comment 15.
Line 427-430 Here also I could not find. The paragraph needs to be shortened and specific points on infertility should be present otherwise no issue of discussing the topic in another section.
Response to Comment 15:
Based on your feedback, we have made modifications to that section. Please refer to lines 519-521. Thank you.
Comment 16.
Comment for table 2 table should be well prepared.
Response to Comment 16:
We have prepared the table more accurately. Please refer to table 2. Your feedback has greatly improved our paper. Thank you.
Comment 17.
Lines 619-620 need to explain more.
Response to Comment 17:
Based on your comment, we have rewritten the portion to provide a clearer description. Please refer to lines 739-747. Thank you.
Comment 18.
General comments for reference list “Only 5 references were found for 2023, I think they should have to work out on the references part as well.
Response to Comment 18:
To the best of our knowledge, we have tried to cite all the relevant references for the year 2023 and added to the manuscript. Please refer to the reference numbers: 3,24,31,72,76,85,142,211,222. Thank you for your insightful feedback.
Reviewer 2 Report
Comments and Suggestions for Authors
According to the manuscript entitled "The Pathological Role of miRNAs in Endometriosis" by Mst Ismat Ara Begum and colleagues. They have reported that association studies investigating miRNA in relation to diseases have consistently shown significant alterations in miRNA expression, particularly within inflammation pathways, where they regulate inflammatory cytokines, transcription factors (such as NF-κB, STAT3, HIF1α), and inflammation-related proteins (including COX-2 and iNOS). It is natural to speculate about the relationship between endometriosis (EMS), which is characterized as an inflammatory disease, albeit one influenced by estrogen levels. Several studies have demonstrated alterations in the expression levels of miRNAs in both endometriotic lesions and eutopic endometriums of women with EMS. It is undeniable that miRNAs are associated with EMS, suggesting the emergence of a new era in miRNA research.In this article, we review the progress made in understanding the pathological role of miRNA in EMS and its association with infertility caused by EMS. MiRNA-based therapeutics and diagnostic markers for EMS are currently being developed based on these findings. Regarding the present manuscript, I would like to make a few comments.
-Authors should follow the journal's template
The manuscript provides novel information, and the rationale of the manuscript is excellent.
-The use of tables and figures is excellent. Adding only one point, they may have to include a summary of the abstract in this journal, but I am not aware of this. To conclude all the work, a graphical abstract may be required.
Author Response
Point-by-point response
Reviewer #2:
Comment 1.
Authors should follow the journal's template.
Response to Comment 1:
We have formatted the manuscript according to the journal template.
Comment 2.
The manuscript provides novel information, and the rationale of the manuscript is excellent.
The use of tables and figures is excellent.
Response to Comment 2:
Thank you so much for your feedback.
Comment 3.
Adding only one point, they may have to include a summary of the abstract in this journal, but I am not aware of this.
Response to Comment 3:
So far, we know that this journal does not require a summary of the abstract. If it is needed, we will submit one. Thank you.
Comment 4.
To conclude all the work, a graphical abstract may be required.
Response to Comment 4:
Up to this point, we are aware that it is a requirement after paper acceptance. Once it is needed, we will submit the graphical abstract of our work. We truly appreciate your comments. Thank you.